# Advanced Statistical Analysis of 3D Kinect Data: Mimetic Muscle Rehabilitation Following Head and Neck Surgeries Causing Facial Paresis

**DOI:** 10.3390/s21010103

**Published:** 2020-12-26

**Authors:** Jan Kohout, Ludmila Verešpejová, Pavel Kříž, Lenka Červená, Karel Štícha, Jan Crha, Kateřina Trnková, Martin Chovanec, Jan Mareš

**Affiliations:** 1Department of Computing and Control Engineering, University of Chemistry and Technology Prague, 1905/5 Technická, 16628 Praha 6, Czech Republic; jan.kohout@vscht.cz (J.K.); karel.sticha@vscht.cz (K.Š.); jan.crha@vscht.cz (J.C.); 2Department of Otorhinolaryngology, 3rd Faculty of Medicine, Charles University Prague, University Hospital Kralovske Vinohrady, 1150/50 Šrobárova, 10034 Praha 10, Czech Republic; ludmila.verespejova@fnkv.cz (L.V.); katerina.trnkova@fnkv.cz (K.T.); martin.chovanec@fnkv.cz (M.C.); 3Department of Mathematics, University of Chemistry and Technology Prague, 1905/5 Technická, 16628 Praha 6, Czech Republic; pavel.kriz@vscht.cz (P.K.); lenka.cervena@vscht.cz (L.Č.)

**Keywords:** rehabilitation, House–Brackmann scale, functional data analysis, ordinal classification, Kinect

## Abstract

An advanced statistical analysis of patients’ faces after specific surgical procedures that temporarily negatively affect the patient’s mimetic muscles is presented. For effective planning of rehabilitation, which typically lasts several months, it is crucial to correctly evaluate the improvement of the mimetic muscle function. The current way of describing the development of rehabilitation depends on the subjective opinion and expertise of the clinician and is not very precise concerning when the most common classification (House–Brackmann scale) is used. Our system is based on a stereovision Kinect camera and an advanced mathematical approach that objectively quantifies the mimetic muscle function independently of the clinician’s opinion. To effectively deal with the complexity of the 3D camera input data and uncertainty of the evaluation process, we designed a three-stage data-analytic procedure combining the calculation of indicators determined by clinicians with advanced statistical methods including functional data analysis and ordinal (multiple) logistic regression. We worked with a dataset of 93 distinct patients and 122 sets of measurements. In comparison to the classification with the House–Brackmann scale the developed system is able to automatically monitor reinnervation of mimetic muscles giving us opportunity to discriminate even small improvements during the course of rehabilitation.

## 1. Introduction

Nowadays, modern medicine is an interdisciplinary field where selected parts of information engineering, cybernetics or signal processing can be found. Typical applications can be seen in early diagnosis, precise surgery (oncology), telemedicine, and rehabilitation. Telerehabilitation in medicine is gaining significant popularity [1] as a very promising tool offering a convenient way to work with patients online. A mobile tablet-based therapy platform for early stroke rehabilitation is described in [2]. A systematic review of Mobile Health Applications in rehabilitation was made by [3]. Assistive technologies for patients are described in [4].

Advanced signal processing plays an important role in biomedical data analysis. The main direction of research in this area is represented by data analysis using neural networks. Neural networks improve brain cancer diagnosis by reducing artifacts [5] or by semi-automatic analysis using a neural network for pattern recognition [6].

Another avenue of recent research has been to employ the power of advanced statistical analysis. Principal component analysis (PCA) combined with linear discriminant analysis was used for the detection of nasopharyngeal cancer [7].

Functional data analysis (FDA) is a dynamically developing modern branch of statistics, which deals with data represented as functions. Such data are typically collected by continuous recording of a certain process, which is also the case of our study (continuous recording of facial expressions during exercises). The FDA method has already reached a certain level of maturity. It provides a rich collection of various methods and techniques including ready-to-use software implementations and has proved to be extremely useful in many diverse fields, including medicine, biomedicine, public health, biology, biomechanics, and environmental science (see [8] for a detailed overview of applications). For more details about FDA in general, see the classical book [9] or the more recent review paper [10].

Regarding analyses of (bio)medical data, FDA has been successfully applied by many researchers. Besides earlier works cited in [8], let us mention the more recent [11] (dimension reduction for functional classification in functional Magnetic Resonance Imaging), [12] (functional ANOVA for analyzing biomechanical gait data), [13] (multivariate functional PCA applied to data from the Alzheimer’s Disease Neuroimaging Initiative study), and [14] (functional LDA applied to relative spinal bone mineral density dataset), to name just a few of them.

### 1.1. Medical Background of the Rehabilitation of Mimetic Muscles

Despite all the advances in medical, surgical, and physical therapy, facial nerve palsy remains a devastating clinical condition with a strong psycho-social and functional influence on the patients. The patients are often plagued with asymmetrical brow position and movement abnormalities, eye closure dysfunction, disturbed oral movement resulting in articulation problems, inability to smile, and facial asymmetry. The loss of tonus of mimetic muscles causes a visible asymmetry of the face, so nonverbal communication also becomes difficult. Patients with facial nerve dysfunction are unable to show their emotions through facial expression. Their emotional state is therefore often misinterpreted [15].

The incidence of peripheral facial palsy ranges from 20–30 cases/100,000/year. It is one of themost common conditions affecting the human cranial nerves. The site of injury can be intracranial, intratemporal, or extracranial. Based on the etiology, we can distinguish traumatic, neoplastic, inflammatory, metabolic, toxic, iatrogenic, congenital, and idiopathic facial nerve palsy.

Facial nerve paresis is generally one of the most feared complications of almost all surgical procedures in head and neck surgery (parotid and submandibular gland surgery, neck dissection, surgery of the middle ear and temporal bone, surgery of the posterolateral skull base).

In patients undergoing surgeries with specific risk of facial nerve injury, damage to the nerve is manifested primarily by disruption of function of mimetic muscle. This leads to either complete paresis or, at best, increased fatigue of the facial muscles, which has a strong influence on the patient’s daily life [15]. An altered oral movement causes problems with articulation, and facial asymmetry results in nonverbal communication problems [16].

Several months of rehabilitation are needed to restore the function of mimetic muscles. The main problem is how to set the rehabilitation to be able to avoid unrequired side effects such as synkinesis. Synkinesis represents unwanted contraction of the muscles of the face during attempted movement, caused by aberrant reinnervation. Commonly, patients notice forceful eye closure when they attempt to smile, or other muscle spasms during routine facial movements [17].

### 1.2. Problems with Evaluation of Facial Nerve Function

Clinical tests and classifications are used to evaluate facial nerve function, as well as electrophysiological methods. The House–Brackmann (HB) classification is probably the most widely employed scale of facial nerve dysfunction that is applied in all fields of clinical medicine. This system carries the name of the Dr. John W. House and Dr. Derald E. Brackmann, american otolaryngologists, who described this system in 1985 (Table 1). The HB scale produces comparable results between different observers in patients with normal or only mildly impaired facial nerve function. However, it has been shown that interobserver variability increased depending on the severity of facial nerve paresis [18].

In patients with variable facial weakness, the single House–Brackmann score does not fully communicate their facial function. The single House–Brackmann score most strongly correlated with the regional scoring of the eye (61%), followed by the nose and midface (40%), mouth (32%), and forehead (18%). The global score does not correlate with the worst regional score. In patients with synkinesis is an obligatory HB3 or higher in the global House–Brackmann grading system, but the regional facial function can be HB2 or better at one or more areas of the face. Furthermore, the single grade does not always correlate with the best or worst function along the four facial regions [20].

There have been significant criticisms of the HB classification, and it is generally agreed that the scale is not effective for determining changes in facial nerve function following a therapeutic intervention. Facial nerve grading systems aim to provide a more uniform and accurate method for assessing facial nerve function. The benefit of using such systems is to allow communication and comparison between practitioners and evaluation of changes in the clinical course. Such a facial nerve grading instrument should document the clinical assessment as objectively as possible and should be sensitive enough to reflect signs of recovery or changes in function following therapeutic intervention (Figure 1). The perfect scale should be: cost effective, fast, minimally invasive, sensitive, specific, objective, and quantitative) [21].

Clinical assessment of facial nerve function is important and will still be part of examination. The main disadvantage is its high degree of subjectivity that has been shown in differences in inter-individual evaluation as well as low usefulness for distinguishing different pathologies that fall into the level of mild functional impairment (e.g., HB3). Furthermore early phases of reinnervation are difficult to assess with clinical examination only. Computer systems can detect slight changes in reinnervation more precisely and compare in time between different clinical sessions. Development of a uniform and accurate method for grading is a prerequisite for effective diagnosis and treatment of patients with facial nerve paralysis [23].

### 1.3. Main Goals of this Article

The main goal of this article is to determine a parametrization that can more objectively describe the rehabilitation process for mimetic muscles by patients after brain surgery with the damage of facial nerve function. There are many classification scales for measuring degrees of facial asymmetries. For the clinical evaluation of mimetic muscle function, the most frequently used is the House–Brackmann scale, which we use for comparison [16].

Due to the wide scope of research and better readability and clarity, we divided the research into two parts: (i) introduction to patient scaling, development of advanced statistical tool and proofs of functionality—this article; (ii) predictive modeling, comparison to other approaches (deep learning) and full (cross) validation—future work.

## 2. Material and Methods

Patients undergoing head and neck surgical procedures with the specific risk for postoperative facial nerve dysfunction or with preexistent facial nerve palsy were enrolled in the study (details in Table 2 and in the related work [24]). There were 93 patients with 122 measurements (one patient was measured over a defined schedule of checkups). The research was approved by the Ethical comitee of University Hospital Královské Vinohrady, Prague (EK-VP/3910120).

### 2.1. Dataset and Measurement Scheme

The measurement scheme is shown in Figure 2. The measurement takes place during checkups, first before the surgery and then repetitively based on a defined schedule. The patient is asked to perform a series of exercises (see Table 3) using the mimetic muscles during the examination, and the clinician evaluates this exercise numerically. A disadvantage, however, is that these measurements are strongly subjective: it depends on the evaluation of clinician during the given examination. However, this assessment strongly influences rehabilitation planning as well as any other actions.

#### 2.1.1. Hardware

Kinect for Windows v2 sensor was chosen as a face data acquisition tool for several reasons: it detects points of the face using software (no physical marks are needed), Microsoft provides an API (Application Programming Interface) which enables object orientated data access, and it is a low-cost solution.

As a multi-sensor device, Kinect v2 includes a color camera, an IR (infrared) sensor and emitters (see Table 4), and a directional microphone array. Depth data are obtained from an active IR sensor by time-of-flight technology [25]. The advantage of this approach is that the resulting image is independent of the illumination of the room. On the other hand, no other sources of IR radiation can be present during measurement due to strong IR radiation interference. Facial features are then extracted by the Microsoft Kinect Face algorithm, which is based on AAM (Active Appearance Model) [26]. The disadvantage is the variable sampling rate of facial features, which may vary from units to tens of Hz. To obtain reliable face data, Kinect v2 is supposed to be preheated to a working temperature, which needs at least 25 min to stabilize [27].

#### 2.1.2. Software

For straightforward face data acquisition, a desktop application was developed. It is designed as a WPF (Windows Presentation Foundation) application for the Windows operating system and is written in the modern programming language C#. Kinect for Windows Runtime 2.0, which is an integral part of the application, provides full control over the Kinect v2 sensor, and the Microsoft Kinect Face library provides APIs for tracking the locations of facial features.

Measured 3D face data, separated for each exercise (see Table 3), are stored in text-based files on which offline analysis can be performed in software like R or MATLAB. As a checkback, the application stores IR records from measurements that can be replayed with 21 mapped tracked facial points—points of interest (POI)—as shown in Table 5.

#### 2.1.3. Mathematical Tools

The choice of an appropriate statistical methodology for the problem considered in this paper was driven by the following three aspects:We have a classification problem with an ordinal response variable (HB grades). In contrast to a multiple response classification, we have ordered responses.The explanatory variables are in the form of (rich) multivariate functional data: 567-time curves in each sample (3 axes × 21 POI × 9 exercises). On the other hand, the sample size is rather limited (122 samples). Hence we faced a severe risk of overfitting.Rather than maximizing the accuracy of the classification, our goal was to design a methodology which would provide reasonable (not perfect) and tractable predictions, including easy-to-understand insight into the main drivers behind the prediction. This is in sharp contrast to a ‘black box’ approach.

The combination of the above three aspects made our statistical problem quite unique. Therefore we developed a three-step methodology combining:calculating of certain indicators,functional linear models for a single logit (see [9]),multivariate linear model for cumulative logits (see [28]).

Due to the high complexity of the input data and so as to make the results easy to understand and interpret, we split the statistical data analysis into the steps that are shown in Figure 3.

### 2.2. Data Preprocessing

In this section, we briefly describe the necessary preprocessing of the data. Each patient performs the exercises at different times and at different speeds. In order to do a further analysis, there is a need for alignment (registration) of the data. Figure 4 illustrates the original data and the desired alignment for the distance between eye corners and eyebrows for three different patients performing Raising (see Table 3).

Let $P_{i}\left(t\right)$ denote the curve representing any coordinate of any facial point for measurement *i*. The curves for all coordinates x,y,z and all facial points for one measurement *i* have to be transformed by a smooth warping function $w_{i}$ so that the exercises are happening at the same instants for all measurements. Thus we are looking for warping functions $w_{i}$ such that the ${\tilde{P}}_{i}$ are aligned for all measurements *i*:(1)$${\tilde{P}}_{i}\left(\tilde{t}\right)=P_{i}\left(w_{i}\left(\tilde{t}\right)\right),where\;\tilde{t}=w_{i}^{-1}\left(t\right)$$

For each exercise, we select one curve that exhibits the most significant changes, such as the distance of the eyebrows from the inner eye corners for Raising, or the distance between the mouth corners for Smiling. Such a curve should include both the left and right facial points since some patients can only move one side of their face. This selected curve will now be used for alignment.

The warping functions $w_{i}$ are computed by a landmark registration technique, where only specific points (landmarks) are aligned. For each measurement, a set of landmarks $t_{ij}$ is identified and the warping function is sought as a piece-wise cubic interpolation function $w_{i}$ satisfying
(2)$$t_{ij}=w_{i}\left(t_{0j}\right),$$
where $t_{0j}$ is a reference set of time instants. Figure 4 illustrates the identified landmarks (beginnings and ends of each repetition) and the piece-wise cubic warping functions for three patients.

The landmarks can be identified manually by looking at the curves, but since there is more data expected in the future, we need an automatic landmark identification. We normalize the curves and align them with a reference curve using dynamic time warping (DTW) in order to identify the beginnings and ends of each repetition of an exercise. The data show that patients perform not always repeat the exercises three times as directed, thus DTW is used also to automatically identify the number of repetitions. For this purpose, the distance between each curve and the reference curves for 2, 3, or 4 repetitions is computed and the one with the minimal average distance is selected, as shown in Figure 5.

Due to the different number of repetitions of an exercise by individual patients, we decided to select a single realization of an exercise for each patient (each measurement). Since the first and the last repetitions are sometimes misidentified (noise at the beginning or at the end of the exercise) we selected the second repetition for each patient. This choice also enables us to reflect on potential tiredness when repeating an exercise.

To avoid distortion of the parametric statistical models by an outlier, we made a visual inspection of the registered curves of indicators. We identified one outlier (a sample with HB6 having enormous eyebrows intensity, smiling intensity, and lips intensity) and excluded it from the data. HB classes 4 and 5 have very low frequencies (compared to the others), this is because these two are rather artificial and used very rarely. HB4 is very similar to HB3, whereas HB5 is similar to HB6. To improve the reliability of the modeling process, we changed the classifications of all samples with HB4 to HB3, and samples with HB5 were reclassified to HB6. The frequencies of the adjusted HB grades are shown in Table 6.

## 3. Results

### 3.1. Indicators Describing Movements of the Mimetic Muscles

For each exercise, we have time-series of *x*, *y*, and *z* coordinates of 21 POI in a face identified by Kinect. To reduce the amount of data and to express the most important properties indicating the rate of facial nerve recovery, we calculated the curves of various indicators (to be specified below) as they changed during a single exercise (Table 3). These indicators measure:symmetry—left half versus right half,intensity—range of motion,speed—how fast the selected exercise is conducted.

The choice of exercises and points of interest is driven mainly by the ability of Kinect to reliably identify the location of the points and their movements, see Figure 6 for illustration. The full list of indicators is shown in Table 7. Figure 7 illustrates differences in selected indicators between fully recovered patients (HB1) and patients with most severe mimetic dysfunction (HB6). Patients with HB1 (recovered) have symmetry indices closer to one, especially for exercises with mouth (smiling, teeth), compared to patients with HB6 (sever mimetic dysfunction). This demonstrates asymmetry between left and right halves of the face for HB6 patients. Regarding intensity indices, patients with HB1 show values farther from zero compared to HB6 patients. This means that the range of motion during exercises is larger for HB1 compared to HB6. Comparison of the shapes of speed indices (warping functions) reveals that patients with HB1 tend to perform the exercise faster than patients with HB6 (warping function is steep at the beginning and flat in the middle for HB1).

#### 3.1.1. Symmetry Indicator

Indicator of symmetry is computed by comparing the left and right distances of two points using the formula
(3)$$SI\left(t\right)=\frac{\min(v\left(P_{L1},P_{L2}\right)\left(t\right),v\left(P_{R1},P_{R2}\right)\left(t\right))}{\max(v\left(P_{L1},P_{L2}\right)\left(t\right),v\left(P_{R1},P_{R2}\right)\left(t\right))}$$
where R1 and R2 denotes two points on the right and L1 and L2 on the left side of the face. For Raising and Frowning we use the distance between the inner eyebrows (points 4 and 10) and inner corners of the eye (points 2 and 8). For Smiling and Baring we use the distance between the outer corners of the mouth (points 14 and 15) and that between the outer corners of the eye (points 3 and 9). Eye corners were chosen as reference points since they were found to be the most stable during the exercises.

#### 3.1.2. Intensity Indicator

Intensity of Raising eyebrows, Intensity of Frowning and Intensity of Smiling is computed from maximum change in left and right distance during the exercise:(4)$$I\left(t\right)=1-\frac{1}{\max\left(\frac{v\left(P_{L1},P_{L2}\right)\left(t\right)}{v\left(P_{L1},P_{L2}\right)\left(0\right)},\frac{v\left(P_{R1},P_{R2}\right)\left(t\right)}{v\left(P_{R1},P_{R2}\right)\left(0\right)}\right)}$$

We consider the same points as the indicator of symmetry. The Intensity of Baring is computed from the change of the area of the ellipse defined by four mouth points (see Figure 6):(5)$$I\left(t\right)=1-\frac{v\left(P_{14},P_{15}\right)\left(0\right)\cdot v\left(P_{13},P_{16}\right)\left(0\right)}{v\left(P_{14},P_{15}\right)\left(t\right)\cdot v\left(P_{13},P_{16}\right)\left(t\right)}$$

The Intensity of Pursing is computed from the change of distance between the corners of the mouth:(6)$$I\left(t\right)=1-\frac{v\left(P_{14},P_{15}\right)\left(0\right)}{v\left(P_{14},P_{15}\right)\left(t\right)}$$

#### 3.1.3. Speed Indicator

For all exercises, we add the warping function (see Section 2.2) as speed indicator.

### 3.2. Health Scores as Descriptors for Mimetic Muscles Function

The next step of our analysis is to reduce the curve of each indicator to a single number (the health score) expressing the rate of recovery (healthiness) for the given exercise and property. Such scores can serve as covariates to determine the final HB grade. Moreover, their evolution in time can help in understanding the process of recovery by identifying improvements (or worsening) in various aspects.

Before the application of Functional Logistic Regression (FLR), we considered two groups of samples (measurements):Healthy—those with HB1,Sick—those with HB5 or HB6.

This enables us to apply the FLR as a functional-data analytic tool which takes the curve (functional datum) of an indicator as a covariate (explanatory variable) and estimates the probability of the sample’s belonging to the Healthy group (response variable) according to the following formula:(7)$$p_{k}=\frac{1}{1+\exp\left(-\alpha -\int X_{k}\left(t\right)\beta \left(t\right)dt\right)},$$
where $p_{k}$ is the probability of the *k*-th sample’s being in the Healthy group, $X_{k}$ the curve of the indicator (a functional covariate) of the *k*-th sample, α the scalar intercept, and β the functional parameter, both being estimated from the data. This model can also be understood as a standard functional linear regression applied to the dependent variable Y=ln(p/(1−p)), which is the logit of *p*. More details about the functional approach can be found in [9]. For our calculations, we made use of an implementation of FLR in the statistical software package R (function classif.glm within the fda.usc package), see [29] for more details.

Having trained the FLR model on the two groups of samples, we used this model to determine the health score (probability of belonging to the Healthy group) for each sample in the dataset (not only HB1, HB5, or HB6). As a result, we obtained a list of scores for all indicators for each sample in the input data.

The application of the FLR to the curves of the calculated indicators provided us with the health scores for each sample and indicator. To assess the relation between these scores and the HB grades (provided by clinicians), we calculated the corresponding paired Spearman’s correlation coefficients (including *p*-values), see Table 8.

We observe moderate or fair (below −0.2) negative Spearman correlation with significant *p*-value (below 0.05) between HB grades and health scores for several indicators. This suggests that (at least some) health scores can serve as useful predictors for the HB classification. The strongest correlation is achieved for symmetries of exercises with the mouth. This is because facial nerve dysfunction is most obvious in this area, both for a clinician and Kinect.

We also studied paired (Pearson’s) correlations between health scores of individual indicators. A strong positive correlation (r=0.71) was observed between smiling.symmetry and teeth.symmetry. Apparently, many patients barely distinguish between Smiling and Baring: they engage similar facial muscles during these two exercises. A weak correlation (below 0.35) but still significant (*p*-value < 0.05) was reported for some other pairs, but these may be spurious correlations. The correlations of most pairs are, however, insignificant. For an overview of the correlations between health scores, see the correlogram in Figure 8.

#### Classification by Ordinal Logistic Regression

The last step of our modeling approach is to classify the samples represented by the lists of health scores into one of the HB grades (HB1, HB2,⋯HB6). In other words, we have a classification problem with a multivariate explanatory variable (list of scores) and an ordinal response variable (HB grade). We applied Ordinal Logistic Regression (OLR)—a parametric statistical method well suited for this kind of problem. This method is an application of a series of standard logistic regressions to cumulative probabilities:(8)$$P\left(HB_{k}\leq j\right)=\frac{1}{1+\exp\left(-\alpha_{j}-\sum_{i}\beta_{i}p_{ik}\right)},$$
with $P(HB_{k}\leq j)$ being the probability of the *k*-th sample’s having HB grade at most *j*, $\alpha_{j}$ the (HB specific) intercept parameter, $\beta_{i}$ the coefficient for the *i*-th health score (independent of the HB level) and $p_{ik}$ the health score for the *i*-th indicator and *k*-th sample. The parameters $\alpha_{j}$ and $\beta_{i}$ are estimated from the data. The probability of a sample’s having a specific HB grade is determined as the difference of cumulative probabilities:(9)$$P\left(HB_{k}=j\right)=P\left(HB_{k}\leq j\right)-P\left(HB_{k}\leq j-1\right).$$

More details of this method can be found in [28] (referred to there as a cumulative logit model). We performed our calculations of OLR using the polr function from the MASS package in R (see [30] for further details).

To avoid overfitting (the HB evaluation includes a subjective judgment of the clinician) and to identify the key properties considered by clinicians when evaluating a patient, we performed a stepwise variable selection procedure based on minimizing the AIC (considering both directions, starting from the empty model).

Furthermore, to compensate for the imbalance of the input data (samples with HB1 dominant) and to avoid overestimation of the probabilities of the dominant class, we applied weights to the individual samples during the fitting process. The weight of a sample was set to the inverse value of the frequency of the corresponding class (HB grade). This weighting procedure balances the accuracy of the classification in individual classes.

The output of the applied OLR method is a set of class probabilities (probabilities of having a specific HB grade) for each sample. The resulting class is then selected as the one with the highest probability. However, the probabilities can be useful in themselves, for example, to quantify the uncertainty of the classification, or to provide a finer characterization of the progress of a single patient.

The health scores obtained by FLR were further used as explanatory variables for HB classification via weighted Ordinal Logistic Regression with stepwise variable selection. We fitted the model on the whole dataset containing 122 valid samples (measurements). The stepwise variable selection procedure (minimizing AIC) selected the following variables (health scores) into the final model (sorted by the order of inclusion into the model):smiling.symmetryteeth.symmetryfrowning.symmetryeyebrows.speedlips.intensityeyebrows.symmetryfrowning.intensitysmiling.speed

The terminology used is shown in Table 9.

A comparison of the graphs in Figure 7 illustrates the differences in the curves of selected indicators between patients with HB1 and patients with HB6. Note that all five considered exercises have been included in the model. Moreover, it contains health scores for symmetry, intensity and speed. This suggests that all important aspects are covered by the model: nothing important has been omitted. On the other hand, the minimization of AIC (which penalizes adding variables to the model) ensures that the final model does not contain any redundant variables. The overall classification performance of the model is summarized in Table 10.

The confusion matrix indicates a reasonable accuracy of classification. The following table summarizes the accuracies of the classification by HB grades and overall accuracy. We considered not only correct classification (HB by clinician = HB by model), but also approximate classification (HB by model differs from HB by the clinician by at most 1).

The rates of correctly classified instances are reasonable, and the rates of approximate classification are satisfactory. Only 14% of the cases were misclassified (Table 11). The weighting of the samples during the fitting process resulted in comparable accuracies between the various HB grades, although the input data are imbalanced (HB1 is dominant).

### 3.3. Case Study

The main objective of our statistical analysis was not to build a model with perfect classification but to build a model which would provide plausible results and would be helpful to a clinician. This means that the results of the model must be easy to understand and interpret. Besides the predicted classes, it has to quantify the uncertainty in the decision. Moreover, it has to provide an insight into the main drivers of the classification and its evolution in time during the rehabilitation process of a patient.

To illustrate the potential for applications of this modeling approach, we include a case study of two selected patients. The first one is a typical patient, and the second one is an untypical patient, who was misclassified by a clinician.

#### 3.3.1. Typical Patient

We have chosen to illustrate the use of our modeling approach on two measurements (sessions) with a single patient having a typical behavior representing the corresponding HB classes (Figure 9).

The patient was evaluated as HB6 in the first session (denoted by Session 1), and HB3 in the other one, which took place 301 days later (denoted by Session 2).

Firstly, let us compare the health scores. The indicators selected for the final OLR model (these affect the final classification) are summarized in Table 12 and illustrated in the following graph.

Figure 10b shows significant improvement of the symmetry of Smiling and Baring. These are the main drivers of the change from HB6 to HB3. The complete asymmetry of these exercises during Session 1 was the main reason for the classification of HB6. Some improvements can also be observed for the symmetry of the eyebrows. The performance during individual exercises in Session 2 was rather balanced, but leaves some room for improvement (therefore HB3).

**Figure 10 sensors-21-00103-f010:**
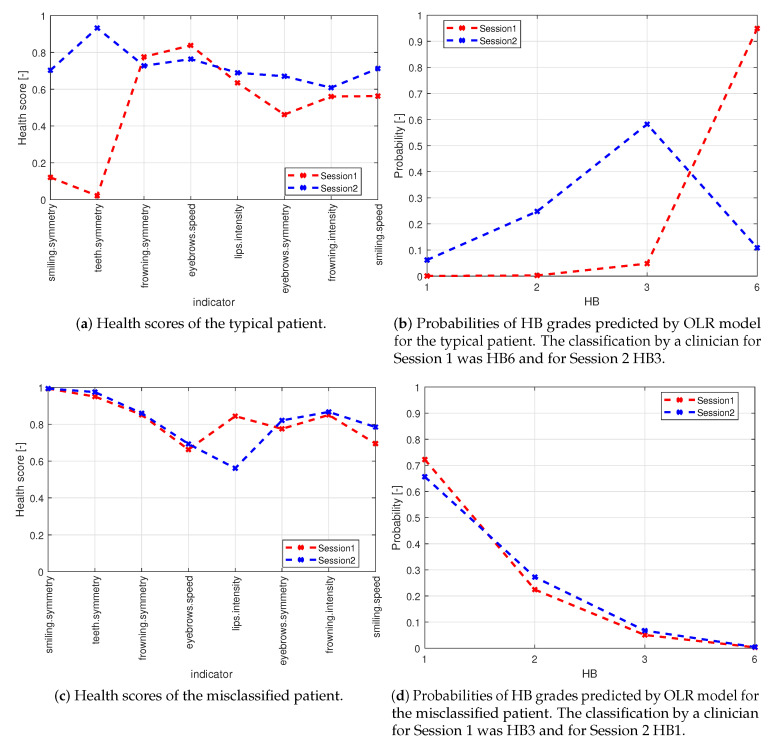
Comparison between typical patient and misclassified patient classified by OLR model. Data shown in Table 13.

**Table 13 sensors-21-00103-t013:** Classification by OLR model for the typical patient.

Session	HB by Clinician	HB by Model	Predicted Probabilities			
			**HB1**	**HB2**	**HB3**	**HB6**
Session 1	6	6	0	0	0.05	0.95
Session 2	3	3	0.06	0.25	0.58	0.11

The predicted HB classes fit perfectly the ones graded by the clinician. This demonstrates the good performance of the model for patients with typical behavior. Moreover, the predicted probabilities of the HB grades tell us that the classification in Session 1 is almost certainly HB6, whereas, in Session 2, most likely (and the chosen one) is HB3, but HB2 has also some not negligible probability. Hence, there is some (but very small) uncertainty in the classification.

#### 3.3.2. Misclassified Patient

We demonstrate the model performance on a patient whose HB classification by a clinician was odd (possibly incorrect). The patient attended two sessions on two consecutive days and obtained HB3 in the first session and HB1 in the second session. However, it usually takes a few months to recover from HB3 to HB1.

Start with health scores for the two sessions summarized in Table 14 and Figure 10c,d.

Apparently, the performance of the patient (expressed in terms of health scores) in the two consecutive days is very similar, which is in line with our expectations. The only non-negligible difference is in the intensity of Pursing, which was better performed during the first session. But this itself should not be a reason to improve HB grade. Moreover, we can see almost perfect symmetry for Smiling and Baring—the main drivers for HB grades. The other indicators can still be slightly improved. We conclude this example with the predictions from the OLR model shown in Table 15.

The model predicted HB1 for both sessions, which is in contrast to the classification by a clinician. However, this should not be understood as a model failure, because the model classification is more reasonable than the odd classification in the input data. Hence, the model corrected an error in the input data. Such advantageous behavior of the model indicates that it is not overfitted, as it fits only reasonable classifications. The probabilities of HB classes determined by the model show that the class HB1 is dominant in both sessions (and therefore chosen); however, there is still some non-negligible probability of HB2. This indicates some possibility of further improvements of the patient, which can potentially be quantified by these probabilities (the six-point HB scale is too coarse).

## 4. Discussion

We developed a three-step statistical procedure for evaluation facial nerve mimetic functions and progress of reinervation following procedures with risk of postoperative palsy. It takes complex multivariate functional data as an input and returns an HB classification as an output. It combines manually designed indicators with advanced statistical techniques including functional data analysis and ordinal logistic regression. We trained the procedure on a sample of 122 measurements (sessions). We see the main advantages of this unique procedure as:its ability to incorporate the experience of the clinicians via calculation of the indicators;providing not only final classification but easy-to-understand insight into the underlying process (quantification of health scores);using modern and advanced statistical methods to extract the maximum information from complex input data;reaching reasonable accuracy in each class, but avoiding overfitting (learning of subjective judgments).

During our statistical analysis, we faced the problem of the enormous noise in the data from Kinect. This made preprocessing the data (smoothing, identification of the beginning and the end of individual repetitions of an exercise, and consecutive registration) extremely difficult. We believe that some important information may have been drowned out by the noise, and lost. We conjecture that this was caused by an unfavorable combination of the uncertainty of the location of some points on the face by Kinect with a rather small intensity (range of motion) of some exercises.

As a result, we had to exclude some exercises from our analysis (such as closing eyes). Moreover, since Kinect works on the basis of a neural network trained on a huge amount of mostly ‘healthy’ (regular, not corrupted by a surgery) faces, it may tend to ‘filter out’ irregularities caused by surgery. In specific, Kinect may tend to locate the points more symmetrically even for patients with severe mimetic dysfunction, which would make them look healthier than they really are. In result, this may complicate the discrimination between HB grades.

## 5. Conclusions

We have introduced a statistical analysis of patients’ faces after specific surgical procedures that temporarily negatively affected the mimetic muscles. We worked with a sample of patients after brain surgery (93 patients, 122 measurements). Our system is based on stereovision data analysis and advanced mathematical methodology that can quantify objectively the degree of mimetic muscle damage, in comparison to the subjective classification carried out by clinicians.

We developed a set of variables: health scores. This set seems to be very promising as a set of objective rehabilitation progress descriptors. In comparison to the HB classification, health scores describe the rehabilitation process more precisely. Base on developed health scores, we created a model for automatic mimetic muscle damage classification. We compared the classification based on our model with that of the clinicians on a case study of selected patients to illustrate the good performance and applicability of the developed model in practical situations.

The three-stage evaluation procedure based on advanced statistical analysis, introduced in this paper, proved to reasonably replicate the subjective evaluation made by clinicians and, in addition, provides the main drivers underlying the evaluation of the progress. Such a system significantly improved the ability to monitor and understand the reasons for the success of the patients’ rehabilitation.

### Future Work

In future work, we will focus more on the predictive modeling of the HB classification. Besides the above parametric models, we plan to test the predictive potential of non-parametric models (such as functional *k*NN in combination with random forests, etc.) and perform the model selection by *n*-fold cross-validation techniques. Moreover, we intend to include possible synkinesis in our models. We would also like to address the problem of the reduction of the noise in the input data from Kinect.

## Figures and Tables

**Figure 1 sensors-21-00103-f001:**
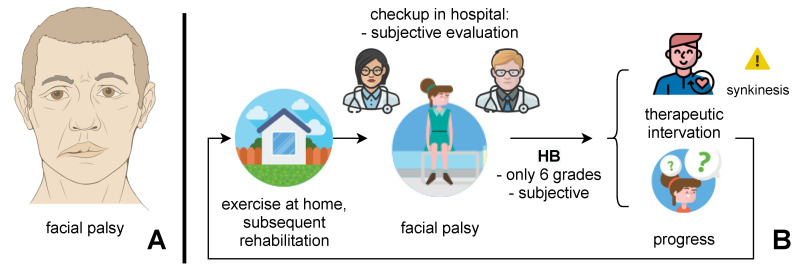
(**A**) illustrates facial palsy. Note the difference between the **left** (sick) and **right** (health) parts of the face. Image source [22]. (**B**) ilustrates problems with clinical classifications, e.g., House–Brackmann classification is observer dependent with six grades only.

**Figure 2 sensors-21-00103-f002:**
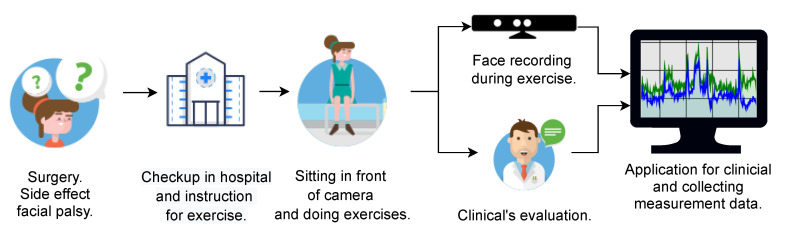
Facial measurement exercise scheme: from surgery to data collecting.

**Figure 3 sensors-21-00103-f003:**
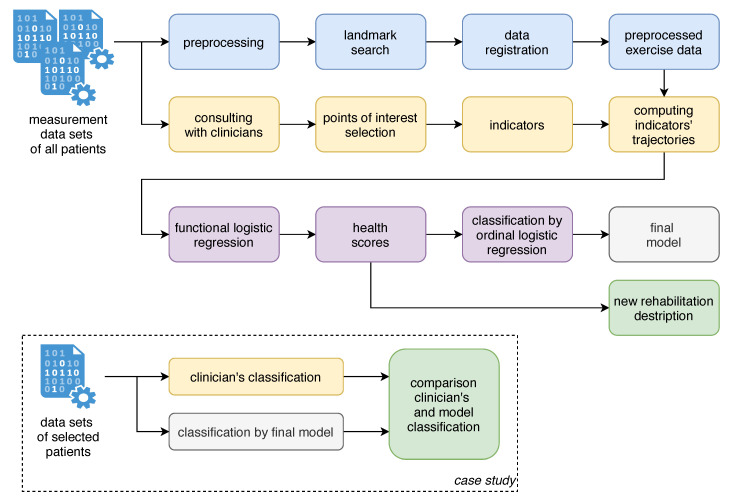
Scheme of analysis steps and main results.

**Figure 4 sensors-21-00103-f004:**
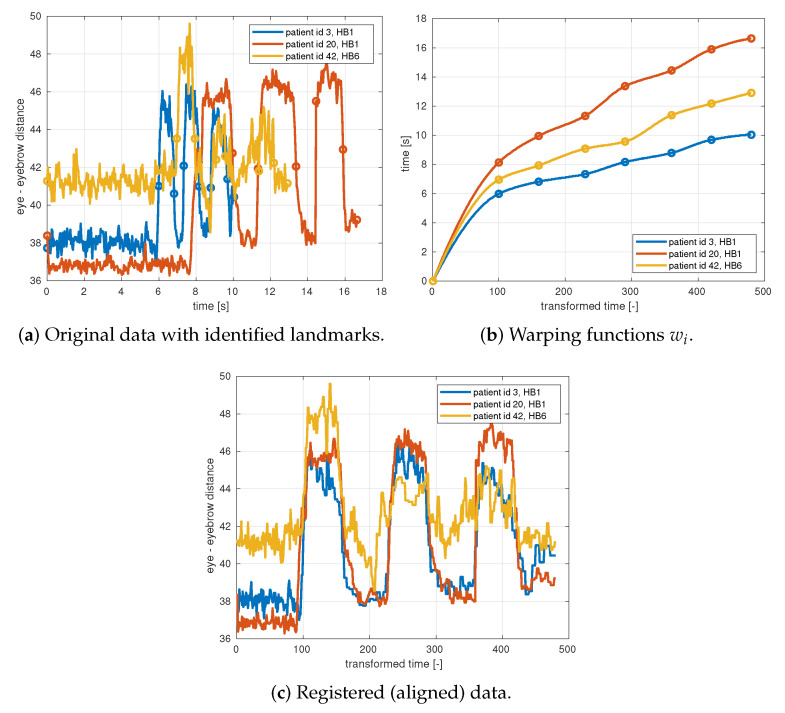
Alignment of data illustrated on the sum of left and right distance between eye corners and eyebrows for three different patients performing Raising exercise.

**Figure 5 sensors-21-00103-f005:**
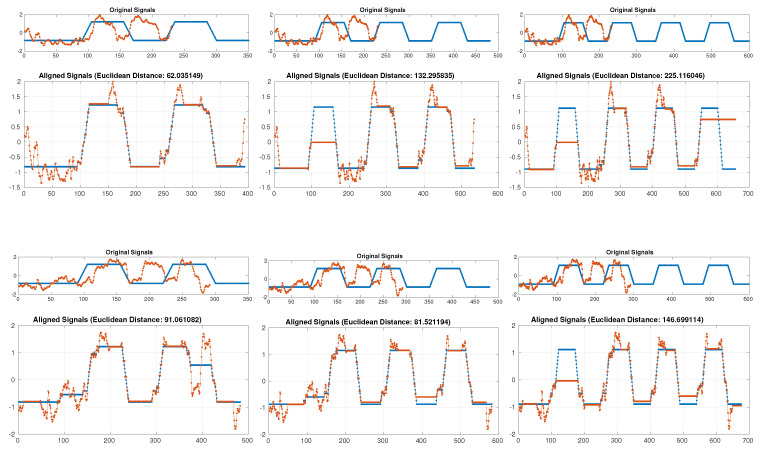
Automatic identification of the number of repetitions. A curve for each patient (red color) is aligned with reference curves for 2, 3 and 4 repetitions (blue). The one with minimal average Euclidean distance is selected. The figure shows Smiling exercise (distance between mouth corners on y-axis, transformed time on x-axis). Two repetitions were selected for the patient with id = 4 (top) and three repetitions for the patient with id = 15.

**Figure 6 sensors-21-00103-f006:**
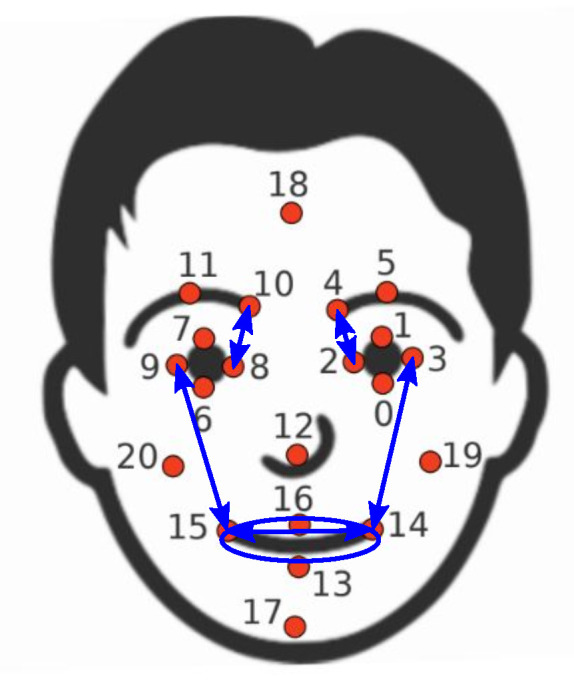
Selected points of interest (POI) (red) and distances (blue) for computation of indicators.

**Figure 7 sensors-21-00103-f007:**
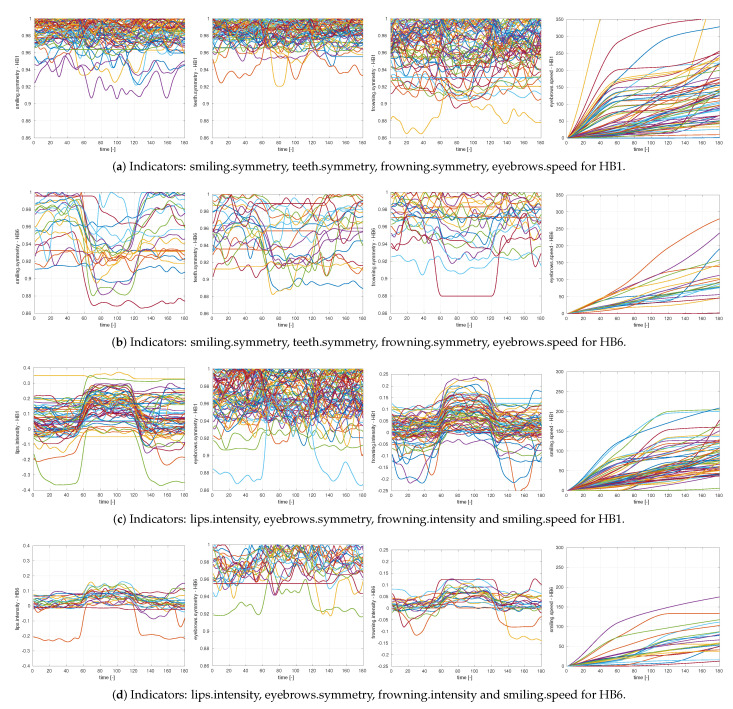
Indicators selected by the stepwise variable selection procedure.

**Figure 8 sensors-21-00103-f008:**
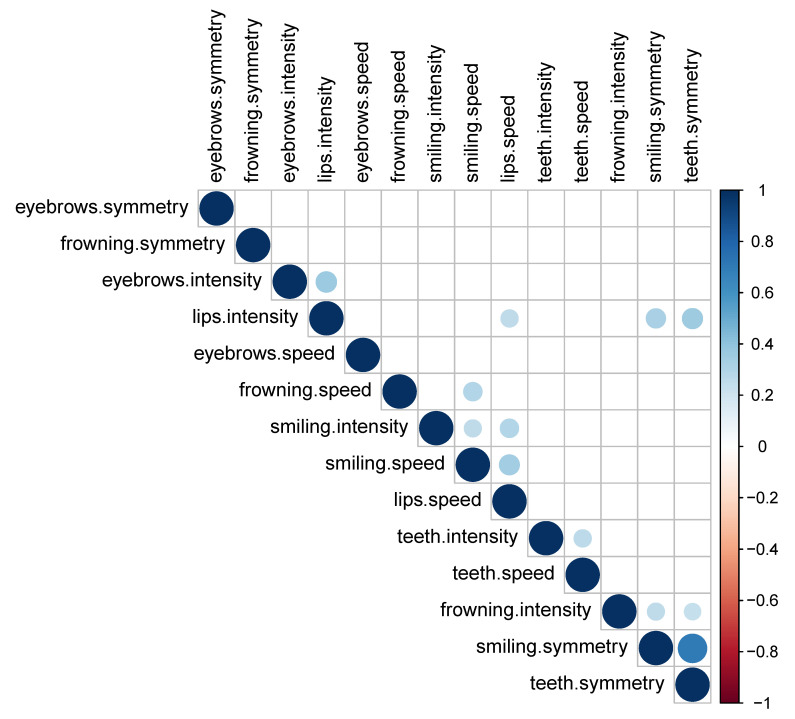
Pearson’s correlogram of health scores for individual indicators.

**Figure 9 sensors-21-00103-f009:**
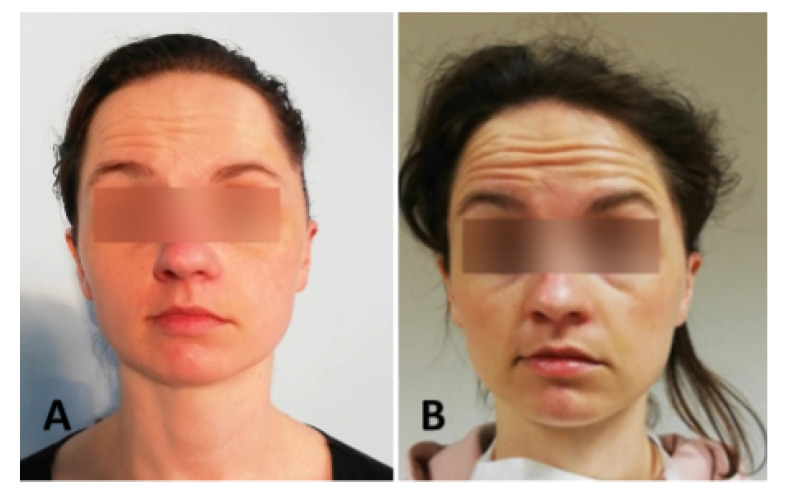
Patient with facial nerve dysfunction raising the eyebrows; (**A**) HB6—no movement on the left side; (**B**) HB3—moderate movement.

**Table 1 sensors-21-00103-t001:** House–Brackmann (HB) classification [19].

Grade	Description	Characteristic
I	Normal function	normal facial function in all areas
II	Mild dysfunction	Gross: slight weakness on close inspection; very slight synkinesis
		At rest: normal tone and symmetry
		Motion Forehead: moderate to good function
		Eye: complete closure with minimum effort
		Mouth: slight asymmetry
III	Moderate dysfunction	Gross: obvious but not disfiguring
		difference between two sides; noticeable synkinesis
		At rest: normal tone and symmetry
		Motion Forehead: slight to moderate movement
		Eye: complete closure with effort
		Mouth: slightly weak with maximum effort
IV	Moderately severe dysfunction	Gross: obvious weakness and disfiguring asymmetry
		At rest: normal tone and symmetry
		Motion Forehead: none
		Eye: incomplete closure
		Mouth: asymmetric with maximum effort
V	Severe dysfunction	Gross: only barely perceptible motion
		At rest: asymmetry
		Motion Forehead: none
		Eye: incomplete closure
		Mouth: slight movement
VI	Total paralysis	no movement

**Table 2 sensors-21-00103-t002:** Dataset: patients.

Gender	
female	52
male	41
average age	57.9
**Diagnosis**	
vestibular schwannoma	55
parotid gland tumor	33
posttraumatic facial nerve palsy	5
**Facial Nerve**	
preoperative palsy	5
postoperative palsy	35

**Table 3 sensors-21-00103-t003:** Table of measured exercises.

Exercise Name	Description for a Patient
Raising	Raise your eyebrows
Frowning	Frown
Closing	Close your eyes tightly
Smiling	Smile at me
Baring	Bare your teeth
Pursing	Purse your lips
Blowing	Blow out your cheeks
Closing and Baring	Close your eyes tightly and bare the teeth
Raising and Pursing	Raise your eyebrows and purse the lips

**Table 4 sensors-21-00103-t004:** Kinect v2 Selected Technical Specification.

Color Camera	
Frame rate—sufficient light	30 fps
Frame rate—insufficient light	15 fps
Resolution	1920×1080
Horizontal × Vertical × Diagonal Viewing Angle	$84.1^{\circ }\times 53.8^{\circ }\times 91.9^{\circ }$
**Infrared and Depth Sensor**	
Measurement Method	Time of Flight
Min.–Max. Distance	0.5–4.5 m
Frame rate	30 fps
Resolution	512×424
Horizontal × Vertical × Diagonal Viewing Angle	$70.6^{\circ }\times 60.0^{\circ }\times 89.5^{\circ }$
**Body and Face Data Sources**	
Face Points	1 347
Face Points Sampling Rate	variable
Number of Cores	2
3.1 GHz	
4 GB	

**Table 5 sensors-21-00103-t005:** Indices of points of interest, *p* is an internal index number.

*p*	Kinect	Position	*p*	Kinect	Position
0	1104	left eye, bottom	11	849	left eyebrow, centre
1	241	left eye, top	12	18	nose tip
2	210	left eye, inner corner	13	8	mouth lower lip, centre-bottom
3	469	left eye, outer corner	14	91	mouth, left corner
4	346	left eyebrow, inner	15	687	mouth, right corner
5	222	left eyebrow, centre	16	19	mouth upper lip, centre-top
6	1090	right eye, bottom	17	4	chin, centre
7	731	right eye, top	18	28	forehead, centre
8	843	right eye, inner	19	412	left cheek, centre
9	1117	right eye, outer	20	933	right cheek, centre
10	803	right eyebrow, inner			

**Table 6 sensors-21-00103-t006:** Frequencies of adjusted HB grades.

	HB1	HB2	HB3	HB6	Total
Nr. of cases	58	21	23	20	122

**Table 7 sensors-21-00103-t007:** Overview of the meanings of the indicators. The position of index *p* is shown in Figure 6.

Exercise	Label	Description
Raising	eyebrows.symmetry	distance between the inner eyebrows (*p* 10 or 4) and inner eye corners (*p* 8 or 2)
Raising	eyebrows.intensity	maximum change in left and right distance
Raising	eyebrows.speed	warping function
Frowning	frowning.symmetry	distance between the inner eyebrows (*p* 10 or 4) and inner eye corners (*p* 8 or 2)
Frowning	frowning.intensity	maximum change in left and right distance
Frowning	frowning.speed	warping function
Smiling	smiling.symmetry	distance between the outer mouth corners (*p* 15 or 14) and outer eye corners (*p* 9 or 3)
Smiling	smiling.intensity	maximum change in left and right distance
Smiling	smiling.speed	warping function
Baring	teeth.symmetry	distance between the outer mouth corners (*p* 15 or 14) and outer eye corners (*p* 9 or 3)
Baring	teeth.intensity	change of the area of the ellipse (*p* 13, 14, 15 and 16)
Baring	teeth.speed	warping function
Pursing	lips.intensity	change of distance between the mouth corners (*p* 14 and 15)
Pursing	lips.speed	warping function

**Table 8 sensors-21-00103-t008:** Paired Spearman’s correlation between House–Brackmann (HB) grades and health scores for individual indicators.

Indicator	Correlation Coefficient	*p*-Value
smiling.symmetry	−0.50	0.00
teeth.symmetry	−0.47	0.00
lips.intensity	−0.40	0.00
frowning.intensity	−0.29	0.00
eyebrows.symmetry	−0.26	0.00
lips.speed	−0.22	0.01
eyebrows.speed	−0.21	0.02
eyebrows.intensity	−0.18	0.04
smiling.intensity	−0.18	0.04
frowning.speed	−0.17	0.06
teeth.intensity	−0.12	0.18
smiling.speed	−0.11	0.21
teeth.speed	−0.09	0.32
frowning.symmetry	−0.08	0.37

**Table 9 sensors-21-00103-t009:** Terminology overview.

Keyword	Description	Values
HB	evaluation of mimetic muscle function (House–Brackmann scale)	1–6
indicator	most important property indicating the rate of facial nerve recovery	string
health score	rate of healthiness for the given exercise and property	single number
trajectory	movement of indicator in time	time series

**Table 10 sensors-21-00103-t010:** Confusion matrix with numbers of (mis)classified cases by Ordinal Logistic Regression (OLR) model.

	HB by Model			
HB by a Clinician	1	2	3	6
1	34	16	8	0
2	7	12	2	0
3	1	7	11	4
6	0	0	4	16

**Table 11 sensors-21-00103-t011:** Accuracy of classification by OLR model.

HB by a Clinician	Correct Classification	Approximate Classification
1	59%	86%
2	57%	100%
3	48%	78%
6	80%	80%
Altogether	60%	86%

**Table 12 sensors-21-00103-t012:** Health scores for two sessions with the typical patient.

	Session 1	Session 2
smiling.symmetry	0.12	0.7
teeth.symmetry	0.02	0.93
frowning.symmetry	0.78	0.73
eyebrows.speed	0.84	0.76
lips.intensity	0.63	0.69
eyebrows.symmetry	0.46	0.67
frowning.intensity	0.56	0.61
smiling.speed	0.56	0.71
HB by clinician	6	3

**Table 14 sensors-21-00103-t014:** Health scores of the misclassified patient.

	Session 1	Session 2
smiling.symmetry	0.99	0.99
teeth.symmetry	0.95	0.97
frowning.symmetry	0.85	0.86
eyebrows.speed	0.66	0.69
lips.intensity	0.84	0.56
eyebrows.symmetry	0.77	0.82
frowning.intensity	0.85	0.87
smiling.speed	0.69	0.78
HB by clinician	3	1

**Table 15 sensors-21-00103-t015:** Classification by OLR model for the misclassified patient.

Session	HB by Clinician	HB by Model	Predicted Probabilities of HB Grades			
			**HB1**	**HB2**	**HB3**	**HB6**
Session 1	3	1	0.72	0.22	0.05	0
Session 2	1	1	0.66	0.27	0.07	0

## Data Availability

The data presented in this study are available on request from the corresponding author. The data are not publicly available due to ethical restrictions.

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
