# Peer review of "Advanced Statistical Analysis of 3D Kinect Data: Mimetic Muscle Rehabilitation Following Head and Neck Surgeries Causing Facial Paresis"

_sensors, 2020, doi:10.3390/s21010103_

Round 1

Reviewer 1 Report

The authors presented a very interesting work that includes current problem in describing the development of rehabilitation after surgical procedures that temporarily negatively affect the patient’s mimetic muscles using their subjective opinion and expertise and common classification (House–Breckman scale).

Here they presented a new system based on a stereovision Kinect camera and an advanced mathematical approach that objectively quantifies the mimetic muscle function independently of the doctor’s opinion. They designed a three-stage data-analytic procedure combining the calculation of indicators determined by the doctors with advanced statistical methods including functional data analysis and ordinal (multiple) logistic regression.

The developed system is able to automatically monitor and describe the rehabilitation, and more precisely than the House–Breckman scale.

Author Response

The authors presented a very interesting work that includes current problem in describing the development of rehabilitation after surgical procedures that temporarily negatively affect the patient’s mimetic muscles using their subjective opinion and expertise and common classification House–Brackmann scale.

Here they presented a new system based on a stereovision Kinect camera and an advanced mathematical approach that objectively quantifies the mimetic muscle function independently of the doctor’s opinion. They designed a three-stage data-analytic procedure combining the calculation of indicators determined by the doctors with advanced statistical methods including functional data analysis and ordinal (multiple) logistic regression.

The developed system is able to automatically monitor and describe the rehabilitation, and more precisely than the House–Brackmann scale.

Answer: Thank you very much for Your positive feedback. We appreciate it. 

Reviewer 2 Report

The paper presents a methodology to describe the development of the rehabilitation after brain surgery causing facial paresis.

The paper is well written and the results are well presented. There are though a few things that are unclear.

The authors have developed a tool for assessing the rehabilitation stage of the post-operatory patients, but the classical medical evaluation is that bad? Why is it necessary to replace the clinician performing this procedure? Or is it a tool for helping the clinician to validate the scores?

Besides, in my opinion there isn’t enough validation of the results. A percentage from the available data (patients/measurements) should be used for a consistent validation.

Several other minor issues:

Section 1, page 3, the authors have listed the HB classification weaknesses. Have they reached these conclusions by themselves? Otherwise citations are needed.

Section 2, page 3: the authors state that their main goal is to provide a “more objective” parametrization for the mimetic muscles of the patients. What does it exactly mean? Are there sufficient data that lead to the fact that doctor’s evaluation is generally wrong?

Please check the English (preferably use “perform” instead of “do” in certain cases).

Author Response

The paper presents a methodology to describe the development of rehabilitation after brain surgery causing facial paresis. The paper is well written and the results are well presented. There are though a few things that are unclear. 

Clinical assessment of facial nerve function

The authors have developed a tool for assessing the rehabilitation stage of the post-operatory patients, but the classical medical evaluation is that bad? Why is it necessary to replace the clinician performing this procedure? Or is it a tool for helping the clinician to validate the scores?

Answer: For better clarity, we added these paragraphs to the text:

Clinical assessment of facial nerve function is important and will still be part of the examination. The big disadvantage is its high degree of subjectivity that has been shown in differences in the inter-individual evaluation as well as low usefulness for distinguishing different pathologies that fall into the level of mild functional impairment (e.g. House-Brackmann Grade 3). Furthermore, early phases of reinnervation are difficult to assess with clinical examination only. Computer systems can detect slight changes in reinnervation more precisely and compare in time between different clinical sessions. 

The development of a uniform and accurate method for grading is a prerequisite for effective diagnosis and treatment of patients with facial nerve paralysis. The House-Brackmann with the clinical examination is extremely helpful but has significant limitations with respect to precision and interobserver reliability. (5)

  1. Brenner MJ, Neely JG. Approaches to grading facial nerve function. Semin Plast Surg. 2004 Feb;18(1):13-22. DOI: 10.1055/s-2004-823119. PMID: 20574466; PMCID: PMC2884698.

Validation

Besides, in my opinion, there isn’t enough validation of the results. A percentage from the available data (patients/measurements) should be used for a consistent validation.

Answer: Thank you for this comment. The research is very complex. Thus we expected to divide it into several parts: (i) introduction to patient scaling, development of advanced statistical tools and proofs of functionality, (ii) predictive modeling, comparison to other approaches (deep learning), and full (cross) validation. Due to the wide scope of research and better readability and clarity, we decided to divide the topic into these two parts.

HB classification weaknesses

Several other minor issues: Section 1, page 3, the authors have listed the HB classification weaknesses. Have they reached these conclusions by themselves? Otherwise, citations are needed.

Answer: Thank you very much for this comment. We added the following paragraphs and references to the manuscript:

The House-Brackmann classification produces comparable results between different observers in patients with normal or only mildly impaired facial nerve function. It has been shown that interobserver variability increased depending on the severity of facial nerve paresis. (2) In patients with variable facial weakness, the single House-Brackmann score does not fully communicate their facial function. The single House-Brackmann score most strongly correlated with the regional scoring of the eye (61%), followed by the nose and midface (40%), mouth (32%), and forehead (18%). The global score does not correlate with the worst regional score. In patients with synkinesis is an obligatory Grade 3 or higher in the global House-Brackmann grading system, but the regional facial function can be Grade 2 or better at one or more areas of the face. Furthermore, the single grade does not always correlate with the best or worst function along the four facial regions.  (3) There have been significant criticisms of the House-Brackmann scale, and it is generally agreed that the scale is not effective for determining changes in facial nerve function following a therapeutic intervention. To improve reporting of outcomes and communication between professionals, consensus and standardization of facial functional assessment are needed. (4)

  1. Scheller C, Wienke A, Tatagiba M, Gharabaghi A, Ramina KF, Scheller K, Prell J, Zenk J, Ganslandt O, Bischoff B, Matthies C, Westermaier T, Antoniadis G, Pedro MT, Rohde V, von Eckardstein K, Kretschmer T, Kornhuber M, Barker FG 2nd, Strauss C. Interobserver variability of the House-Brackmann facial nerve grading system for the analysis of a randomized multi-center phase III trial. Acta Neurochir (Wien). 2017 Apr;159(4):733-738. doi: 10.1007/s00701-017-3109-0. Epub 2017 Feb 10. PMID: 28188418.
  2. Yen TL, Driscoll CL, Lalwani AK. Significance of House-Brackmann facial nerve grading global score in the setting of differential facial nerve function. Otol Neurotol. 2003 Jan;24(1):118-22. DOI: 10.1097/00129492-200301000-00023. PMID: 12544040.
  3. Fattah AY, Gurusinghe AD, Gavilan J, Hadlock TA, Marcus JR, Marres H, Nduka CC, Slattery WH, Snyder-Warwick AK; Sir Charles Bell Society. Facial nerve grading instruments: a systematic review of the literature and suggestion for uniformity. Plast Reconstr Surg. 2015 Feb;135(2):569-79. DOI: 10.1097/PRS.0000000000000905. PMID: 25357164.

More objective” parametrization explanation

Section 2, page 3: the authors state that their main goal is to provide a “more objective” parametrization for the mimetic muscles of the patients. What does it exactly mean? Are there sufficient data that lead to the fact that a doctor’s evaluation is generally wrong? 

Answer: Thanks for this question. We added this paragraph to the manuscript:

Facial nerve grading systems aim to provide a more uniform and accurate method for assessing facial nerve function. The benefit of using such systems is to allow communication and comparison between practitioners and evaluation of changes in the clinical course of facial nerve dysfunction and its reinnervation. Such a facial nerve grading instrument should document the clinical assessment as objectively as possible and should be sensitive enough to reflect signs of recovery or changes in function following therapeutic intervention. The perfect scale should be (i.e., cost-effective, fast, minimally invasive, sensitive, specific, objective, and quantitative). (4)

(4) Fattah AY, Gurusinghe AD, Gavilan J, Hadlock TA, Marcus JR, Marres H, Nduka CC, Slattery WH, Snyder-Warwick AK; Sir Charles Bell Society. Facial nerve grading instruments: a systematic review of the literature and suggestion for uniformity. Plast Reconstr Surg. 2015 Feb;135(2):569-79. DOI: 10.1097/PRS.0000000000000905. PMID: 25357164.

English check

Please check the English (preferably use “perform” instead of “do” in certain cases). 

Answer: Done (2x) in Section 2.1, Section 2.2. Professional proof-reading was performed.

Reviewer 3 Report

This paper introduces a novel method for estimating the mimetic muscle affectation in patients suffering from facial paresis. The main concept behind this approach is the use of a Kinect device for capturing the patients’ facial interest points and computing the damage level using symmetry, intensity and speed criteria. Values are statistically processed to determine the House Brackmann (HB) score, which is the standard measure in the field. Results show that the proposed approach can actually assist doctors in this task.

Overall, this paper addresses an interesting topic on statistical tools for biomedical image classification. The paper is well written; it is easy to read, and to follow. I consider it a solid contribution to Sensors.  Still, I’m recommending a major revision with the following remarks:

1) The HB classification is not given. After reading the paper, non-related readers to this term assume that it is a classification for facial nerve damage and that it goes from HB1 to HB6. What are the main characteristics that define each of the six HBs? In addition, Figure 6 needs a more extensive discussion to make clearer such HB level comparison.

2) Please include the explanation for the term synkinesis 

3) In Table 6, include the points concerned with each label.

4) Please consider including a facial image of one of your two case studies being careful of respecting privacy (example: occluding the eyes or showing only a specific area such as the mouth) to have an example of how your symmetry, intensity, and speed values are obtained.

5) Last, lines 339-341, mentioning the Kinect undesired facial filtering is disturbing. How is this affecting your application? No details are given.

Grammar:

6) Line 55: most common conditions to affect the human …

--> most common conditions affecting the human… 

Author Response

This paper introduces a novel method for estimating the mimetic muscle affectation in patients suffering from facial paresis. The main concept behind this approach is the use of a Kinect device for capturing the patients’ facial interest points and computing the damage level using symmetry, intensity, and speed criteria. Values are statistically processed to determine the House Brackmann (HB) score, which is the standard measure in the field. Results show that the proposed approach can actually assist doctors in this task.

Overall, this paper addresses an interesting topic on statistical tools for biomedical image classification. The paper is well written; it is easy to read and to follow. I consider it a solid contribution to Sensors.  Still, I’m recommending a major revision with the following remarks:

HB classification

1) The HB classification is not given. After reading the paper, non-related readers to this term assume that it is a classification for facial nerve damage and that it goes from HB1 to HB6. What are the main characteristics that define each of the six HBs?

Answer: Thank you very much for this comment. We added Table 1. House-Brackmann scale (HB) and the following paragraphs and references to the manuscript:

House-Brackmann classification is probably the most widely employed scale of facial nerve dysfunction that is applied in all fields of clinical medicine. This system carries the name of Dr. John W House and Dr. Derald E. Brackmann, American otolaryngologists, who described this system in 1985. For readers not aware of this system we added the full description of all 6 grades. (1) 

  1. House JW, Brackmann DE. Facial nerve grading system. Otolaryngol Head Neck Surg. 1985 Apr;93(2):146-7. DOI: 10.1177/019459988509300202. PMID: 3921901.

Clarification of Figure 6

In addition, Figure 6 needs a more extensive discussion to make clearer such HB level comparison. 

Answer: We appreciate this comment. The better discussion was added to Sec. 3.1

Synkinesis

2) Please include the explanation for the term synkinesis  

Answer: Thank you. We added this paragraph into Sec. 1.1 for better clarity: 

Synkinesis represents an unwanted contraction of the muscles of the face during attempted movement, caused by aberrant reinnervation.  Commonly, patients notice forceful eye closure when they attempt to smile, or other muscle spasms during routine facial movements. 

Table 6 with labels

3) In Table 6, include the points concerned with each label. 

Answer: Thank you for the comment, points concerned with each label were included (added to Tab. 6).

Facial image

4) Please consider including a facial image of one of your two case studies being careful of respecting privacy (example: occluding the eyes or showing only a specific area such as the mouth) to have an example of how your symmetry, intensity, and speed values are obtained. 

Answer: Thank you for this comment. We have included in the article a patient with facial nerve dysfunction. The patient in this case is raising the eyebrows, in picture A, you can see asymmetry, no wrinkles, and any sign of movements,  in picture B, is noticed the beginning of the symmetry movements, and also the wrinkles. Added into Figure 9.

Kinect facial filtering

5) Last, lines 339-341, mentioning the Kinect undesired facial filtering is disturbing. How is this affecting your application? No details are given. 

Answer: Thank you for this note. Added to the end of Sec. 4 at the end of the last paragraph:

In specific, Kinect may tend to locate the points more symmetrically even for patients with severe mimetic dysfunction, which would make them look healthier than they really are. As a result, this may complicate the discrimination between HB grades.

English

6) Line 55: most common conditions to affect the human … --> most common conditions affecting the human… 

Answer: Corrected. Professional proof-reading was performed.

Round 2

Reviewer 2 Report

The authors have submitted and eplained well the issues I've stressed in previous review process. Please check again for minor typos

Reviewer 3 Report

The manuscript has undoubtedly improved from its previous version. I appreciate that my comments and suggestions were taken into account. The authors definitively made a good work answering my remarks and making the corresponding modifications. In particular, the House–Brackmann 
classification is now clear for non-specialist readers and the new Figure 9 effectively shows the nerve dysfunction in real patients.

The paper is a solid contribution to Sensors. I have no further comments; therefore I recommend now its acceptance.